# Multi-Step Tail Biting Outbreak Intervention Protocols for Pigs Housed on Slatted Floors

**DOI:** 10.3390/ani9080582

**Published:** 2019-08-20

**Authors:** Jen-Yun Chou, Keelin O’Driscoll, Rick B. D’Eath, Dale A. Sandercock, Irene Camerlink

**Affiliations:** 1Pig Development Department, Animal & Grassland Research and Innovation Centre, Teagasc, P61 P302 Moorepark, Ireland; 2Animal & Veterinary Sciences Research Group, SRUC, Roslin Institute Building, Easter Bush, Midlothian EH25 9RG, UK; 3Royal (Dick) School of Veterinary Studies, University of Edinburgh, Easter Bush, Midlothian EH25 9RG, UK; 4Institute of Animal Welfare Science, University of Veterinary Medicine, Veterinärplatz 1, 1210 Vienna, Austria

**Keywords:** undocked tail, tail docking, tail biting, fully slatted floor, victim, enrichment, tail score, pig

## Abstract

**Simple Summary:**

Tail biting is an unpredictable and costly damaging behaviour among pigs that causes painful injuries. A major concern of the industry is how to control tail biting outbreaks when they occur. We aimed to evaluate the effectiveness of three different interventions to overcome tail biting outbreaks: removing biter, removing victim, or providing three ropes; deployed in random order. If the first intervention failed, a second was used, and then a third if that also failed. Data were collected from two studies in which a total of 1248 pigs (96 pens) were housed on fully-slatted floors. Forty tail biting outbreaks were recorded, of which twenty were resolved using only one intervention. Eighty percent of all outbreaks were controlled within three intervention steps. Adding ropes was the fastest way to stop an outbreak but did not stop more than the other strategies; success depended more on the proportion of tail biting pigs in the pen than the intervention used. Removed victims and biters were successfully reintroduced back to the original group, following set rules. This is crucial to reduce the need for space and renders the interventions applicable on commercial farms. It is promising that most outbreaks were overcome using these cost-effective measures.

**Abstract:**

Solutions are needed to keep pigs under commercial conditions without tail biting outbreaks (TBOs). However, as TBOs are inevitable, even in well managed farms, it is crucial to know how to manage TBOs when they occur. We evaluated the effectiveness of multi-step intervention protocols to control TBOs. Across 96 pens (1248 undocked pigs) managed on fully-slatted floors, 40 TBOs were recorded (≥3 out of 12–14 pigs with fresh tail wounds). When an outbreak was identified, either the biters or the victims were removed, or enrichment (three ropes) was added. If the intervention failed, another intervention was randomly used until all three interventions had been deployed once. Fifty percent of TBOs were controlled after one intervention, 30% after 2–3 interventions, and 20% remained uncontrolled. A high proportion of biters/victims per pen reduced intervention success more so than the type of intervention. When only one intervention was used, adding ropes was the fastest method to overcome TBOs. Removed biters and victims were successfully reintroduced within 14 days back to their home pens. In conclusion, 80% of TBOs were successfully controlled within 18.4 ± 1.7 days on average using one or multiple cost-effective intervention strategies.

## 1. Introduction

Tail biting in commercially reared pigs is an injurious behaviour that is one of the main animal welfare issues in the pig sector. Despite routine tail docking being banned in the EU [1], more than 95% of pigs are tail-docked shortly after birth to reduce tail biting using an exemption from the regulation [2]. The exemption is allowed when farmers demonstrate that tail biting on their farm cannot be properly controlled by other means than tail docking [3]. However, with increasing pressure from the public and governments to stop tail docking [4], an expanding body of research has focused on finding solutions to reduce, predict, and prevent the occurrence of tail biting amongst undocked pigs [5,6,7,8].

Due to the multifactorial nature of tail biting behaviour, it will likely prove impossible to completely eradicate it; even in outdoor pig herds, there have been records of tail biting, albeit with a much lower risk [9]. Severely injurious tail biting behaviour can quickly spread from a single pig to its pen mates and even to other pigs in adjacent pens [10]. This kind of rapid development and contagion is defined as a tail biting outbreak [11] and is associated with considerable economic losses [3]. It is, therefore, crucial to investigate intervention strategies to control tail biting outbreaks when they occur to reduce the negative consequences for farm economics and animal welfare. 

To date, most of the advice that farmers receive regarding different tail biting outbreak interventions is based on experience and anecdotes, and there are few scientific studies that specifically evaluate intervention effectiveness. This is primarily because tail biting outbreaks are unpredictable in nature, which creates difficulty in terms of data collection [5]. There have been attempts to stimulate tail biting outbreaks for research purposes, but these often fail to induce the behaviour [12]. Bracke used a rope model as surrogate to tail biting and showed that, when ropes were covered with either Stockholm tar or Dippel’s oil, pigs’ rope manipulation reduced. Therefore, these two substances may be used to halt tail biting [13]. However, pigs’ reaction to a synthetic rope and an actual pig’s tail is likely to be different, especially in the context of a tail biting outbreak. Moreover, for ethical reasons, tail biting occurrences in research settings are treated early on to minimize harm inflicted on experimental animals. For instance, in a pioneering study of tail biting outbreak management, an outbreak was defined as occurring when one pig had a fresh tail wound and one other pig had a tail wound/bite mark out of a pen of 10 pigs [14]. Later, Lahrmann et al. classified tail biting outbreaks using detailed examination of the tail for lesions to study early intervention strategies [15]. In their study, when four out of 30 pigs/pen had tail lesions, regardless of severity, these pens were classified as outbreak pens [15]. The assessment of how effective the interventions would be was dependent on the definitions given—whether at an earlier or a later stage of development. Another study classified outbreaks in a more detailed fashion into two categories; an *underlying* outbreak was detected by detailed tail examination, and a *severe* outbreak was detected when blood and severe damage could be seen on two out of 30 pigs (without close-up examination) [16]. The severe outbreaks were the ones that included clinical signs, but the damage observed by the authors was still quite varied [16]. In practice, it is unlikely that farmers will enter the pens to assess tail damage in individual pigs. Entering the pen is also not recommended from the perspective of disease transmission. Thus, recent studies on early detection of tail biting have begun to adopt a more commercially relevant definition of an outbreak based on pen-side recordings to improve the transferability of results to commercial practice [17,18]. 

Surveys have shown that farmers from different countries in Europe have different attitudes on how to handle tail biting outbreaks. Dutch pig farmers prefer to remove both tail biters and victims [19]. Swedish farmers intervene by removing biters, followed by victims, then increasing straw rations [20]. Farmers in Finland reported using a variety of intervention methods including removing biters, adding more bedding materials, removing victims, and reducing stocking density [21]. However, in England and Ireland, surveys have shown that farmers mainly tend to remove the victims as the primary solution [22,23]. In many instances, removing animals from the group was considered an important step to control tail biting, but continuous removal of animals can create issues due to limited availability of hospital pens on farms [5,21]. The removed pigs are rarely reintroduced to avoid aggression [24] and hence create an additional space requirement with extra cost and management. Zonderland et al. (2008) had to regroup the removed biters into new pens constantly to limit space use but made no mention of the regrouping methods and measures to reduce aggression at mixing [14]. Based on current research recommendations, provision of loose straw can stop a tail biting outbreak effectively and efficiently [14,15], even with only 10 g of long straw per pig twice a day. In housing with partly or fully slatted floors, however, the provision of straw can block the slurry system [5], even when only as little as 5 g/pig/day of straw is provided [14]. There is, therefore, a need for investigation of practical measures to tackle tail biting outbreaks that are compatible with slatted floor systems, which still predominate in pig production. 

We aimed to evaluate the effectiveness of different intervention methods incorporated in a multi-step protocol to control tail biting outbreaks in pigs under commercial housing conditions. Tail biting outbreak data were collected from pigs housed on fully-slatted floors. Outbreaks were recorded according to a clear definition and a threshold using a level of tail damage that could easily be seen from outside the pen, which we believe is similar to that used in commercial practice. Three different intervention methods suitable for use in practice were randomly assigned to pens with an outbreak: removing the biters, removing the victims, or adding additional enrichment in addition to material already present. The intervention was classified as successful only when the removed biters and/or victims could be reintroduced into the original group without further tail biting. This study thereby addresses two aims: to compare practical intervention strategies during tail biting outbreaks and to evaluate the possibility of reintroduction of removed pigs back to the group. 

## 2. Materials and Methods

Data were collected from two similar research trials conducted at the Teagasc Pig Research Facility, Moorepark, Ireland, both investigating tail biting in undocked pigs. The first trial studied the effect of dietary fibre and a single environmental enrichment (for details, see Chou et al. (2019) [25]), and the second looked at the effect of different enrichment management strategies (for details, see Chou et al. (2019) [26]). Both trials were conducted at the same research facility using the same pig herd and managed with identical husbandry practices (housing, ventilation, and lighting). However, the major difference between two trials was the complexity of environmental enrichment used. Pigs in the second trial were housed in a more enriched environment (eight different point-source enrichment items including an elevated rack with grass per 12 pigs) compared to the first trial (one point-source item per 14 pigs). Specific steps in data analyses were taken to account for this variation.

The trials were approved by the Teagasc Animal Ethics Committee (TAEC124/2016 and TAEC163-2017).

### 2.1. Animals and Housing

Across 96 groups (48 groups in each trial), 1248 male and female pigs (Large White × Landrace) were studied from weaning to slaughter (from 5 to 20 weeks of age). Pigs were reared according to conventional commercial practice in Ireland where male piglets are not castrated. Pigs’ tails were left intact (not docked), but their needle teeth tips were clipped at 1 day of age to reduce skin lesions and damage to sows’ udders. Groups consisted of 12 to 14 pigs and were of mixed sex (50/50 male/female ratio). The weaner pens measured 2.4 × 2.6 m and fattener pens 4 × 2.4 m, and both had fully slatted floors. In the weaning housing, the temperature was kept at around 28 °C immediately post-weaning and reduced by 2 °C every 2 weeks thereafter, while in the fattening housing, the temperature was maintained at 20 °C. Lighting in the weaner house was provided by artificial lighting at 150 lux and at 130 Lux in the finisher house, from 8:00 to 18:00, with windows along the walls providing natural daylight. The pigs had ad libitum access to water via a nipple drinker and dry pelleted feed from a single space wet-dry feeder (standard weaner feed: metabolisable energy 13.7 MJ/kg, crude protein 196 g/kg, crude fibre 37 g/kg; standard finisher feed: metabolisable energy 12.5 MJ/kg, crude protein 142 g/kg, crude fibre 59 g/kg). At 11 weeks of age, the pigs were transferred to the finisher housing without further regrouping. Spare empty pens of the same dimensions, which were in the same room or building and under the same management as the experimental pens, were used as the hospital pens. The hospital pens had point-source items available as enrichment. 

### 2.2. Recording of Tail Injuries and Tail Scores

Pigs were individually identified by ear tags. Routine health checks were made by the experimenters three times daily (9:00, 14:00, and 17:00, when the pigs were most active) and by the farm staff at different time points throughout the day (9:00–12:00 and 14:00–16:00). All inspections were conducted from outside the pen. At each of these health checks, pigs’ tail were inspected using a visual guide to look for swollen tail wounds and fresh dripping blood (maximum damage/blood on the scoring system developed by the FareWellDock consortium [27], Appendix A
Figure A1). Pigs observed to be actively tail biting were also identified and recorded as a “biter” using their ear tag number. As per the experimental protocols of the main studies, more detailed tail lesion scoring on every individual pig was conducted every two weeks. This was carried out within the pen by a single recorder using the same scoring system.

### 2.3. Definition of a Tail Biting Outbreak

A tail biting outbreak within a pen was defined as (a) 3 or more pigs in a pen (of 12–14 pigs, i.e., 21.4–25%) with fresh dripping blood (blood score 3 of the FareWellDock system) present on their tails, clearly visible from outside the pen, (b) 1–2 pigs (not necessarily the same pigs) with fresh bloody tails in a pen, but for 72 h, or (c) 3 or more pigs with tail damage score 3 (of the FareWellDock system) for 72 h, but fresh blood not present. Instances of (a) were further classified as acute outbreaks, whereas (b) and (c), i.e., lasting ≥ 72 h, were considered slow outbreaks.

### 2.4. Intervention Methods

Three intervention methods were used:

1. Removing victims: Victims, identified through the methods described above, were removed and treated with topical antibiotic spray (Alamycin^®^ Aerosol, Norbrook, Newry, Northern Ireland) on the tail and rear area to prevent infection of affected tails. Dettol spray (Dettol, Reckitt Benckiser, Slough, UK, diluted at 1:20 ratio) was also used to reduce the scent and the attention towards Alamycin^®^. Removed pigs were housed by original pen in separate hospital pens and therefore not mixed with others.

2. Removing biter(s): Once an outbreak occurred, biters were identified through 10–15 min of behaviour observation, examination of tails (tail-biters often do not have as much tail damage as pigs that are bitten), and previous tail biting history. When only one biter was identified, it was removed with another non-bitten pig (having uninjured full-length tails, or if there were none, a litter-mate with the lowest tail injury score) to reduce stress while away from the home pen and facilitate later reintroduction. Removed pigs were not mixed with pigs from other pens or with removed victims (as the intervention methods were not applied simultaneously for a pen). On removal of the biter(s) and any companion pig, all the victim pigs remaining in the pen were treated with Alamycin^®^ spray (Alamycin^®^ Aerosol) on the tail and rear area to prevent infection of affected tails. A topical spray without antibiotic (Repiderma, Intracare, The Netherlands) was applied to non-victims, and Dettol spray was also used to divert pigs’ attention from the victim. 

3. Adding 3 ropes: Three 1 m synthetic hemp ropes (without knots) were provided by hanging them from three different sides of the pen. An ointment (Cheno Unction, PharVet, Ireland) was applied on all tails and Dettol spray on the tail and rear area to prevent infection and reduce the scent of blood. The ointment was reapplied whenever necessary. 

### 2.5. Intervention Protocol

This study employed a multi-step intervention protocol using the three intervention methods described above (Figure 1). The order of the three intervention methods was assigned randomly for each outbreak pen without repetition within an outbreak. When a tail biting outbreak was identified, the first intervention out of the predetermined and randomised set of protocols was applied. From the next routine inspection onwards, if fresh blood was observed on the same victim(s) or new victim(s) within 72 h after intervention, the method was regarded as having failed, and a second randomly selected intervention was deployed. An intervention was considered successful when no fresh blood was observed within 72 h.

Up to a maximum of 3 interventions were deployed per pen. If all intervention methods failed and tail biting still could not be controlled, other additional measures were taken (e.g., provision of more enrichment such as feed bags, quarantine of more pigs, application of ointment on tails, etc.). If the intervention was successful, the removed pigs (biters or victims) were reintroduced after the 72 h period. Upon reintroduction, three ropes were provided, and diluted antiseptic spray (Dettol, diluted at 1:20 ratio) was applied in the pen to minimise aggression, as this was a standard practice on farm. In the next 72 h, one rope was removed per day if no further biting occurred. After all ropes were removed, if no fresh blood was observed in another 72 h, the reintroduction procedure was complete, and the outbreak was deemed resolved (Table 1).

If the intervention was “adding rope”, then after the intervention was successful, one rope was removed per 24 h until all three ropes were removed from the pen (within 72 h). Similarly, as previously described, if no fresh blood was observed in the next 72 h, the outbreak was deemed resolved (Table 1).

Removed pigs were never mixed with pigs unfamiliar to them. In order to facilitate reintroduction of the removed animals, besides using ropes and antiseptic spray as distractions as described above, the following rules were applied: (a) at least two pigs were removed together and later reintroduced together, (b) pigs were always reintroduced to the group where they came from, and (c) pigs were returned within 14 days of removal, which is a cut-off point decided in advance to minimise possible aggression. Within 72 h after each outbreak took place, any recurring outbreak was regarded as the same outbreak, and that intervention method was considered as having failed. Therefore, the next intervention was deployed instead of being treated as a new outbreak. 

In trial 1, the choice of the intervention was fully randomised without replacement. Because this led to an overrepresented group in which the biter was removed as the first intervention (50%; 13 out of 26 first interventions), in trial 2, the interventions were randomised, but the first intervention method was controlled to best balance the frequency of all three methods to 1/3. The implemented protocols were clearly recorded and signposted on treatment pens so that no other procedures were taken to confound the effects. 

### 2.6. Ethical Considerations

If, at any stage, tail damage was severe (e.g., suspected amputation of part of the tail or severe bites and inflammation), an injection of antibiotic (Norocillin^®^, Norbrook, Newry, Northern Ireland) and analgesic (Loxicom^®^, Norbrook, Newry, Northern Ireland) was administered to the affected pig. If, after an intervention, bitten pigs had scabs or tail wounds that were a risk of infection, then reintroduction or removal of ropes was postponed for ethical reasons. In pens without reaching the criteria of a tail biting outbreak, removal of tail bitten victims was carried out using the same protocol as for outbreaks (removal with a companion and reintroduction) whenever appropriate, and medical treatment was applied to the tails as part of the main studies’ procedures. If pigs had prolonged tail infection without recovery or if the tail injury affected their overall health and welfare, as evaluated by the experimenter and the staff, then they were removed from the study, and euthanasia was practiced as a humane end point when necessary.

### 2.7. Data Analyses

Data were analysed in SAS version 9.4 with each outbreak occurrence as the statistical unit. If the same pen had more than one outbreak, only the data of the first outbreak per pen were analysed together. As there were only six out of 34 pens that had a second outbreak, these data were described by descriptive statistics. 

Differences between the trials were compared using the Chi-square goodness of fit test for the number of pens with outbreaks, the number of pens with slow outbreaks, and the number of successful interventions. A 2-sample t-test was used to compare the mean duration of outbreaks, and the Mann–Whitney test was used for number of interventions used (not normally distributed). The course of the outbreaks per intervention protocol was analysed using Kaplan–Meier survival analysis with the Lifetest procedure. The pair-wise comparisons between the protocols were done by paired testing using log-rank test. 

To assess the effect of intervention method on the control of tail biting outbreaks, the duration of the outbreaks and the proportion of pigs with tail score 0, 2–3, or 3 were analysed using general linear mixed models (mixed procedure), whereas the intervention success was analysed using a logistic model (GLIMMIX) using the binary distribution and the logit link function. In both models, batch (i.e., replicate) nested within trial was included as random effect.

In both models, the initial explanatory variables were trial (1, 2), month of the year (12 months), first or last method applied (biter(s) removed (B), victim(s) removed (V), ropes given (R)), number of interventions used (1, 2, 3), interaction between the last method × number of interventions, proportion of biters and victims identified during the course of the outbreak, week of age at time of outbreak, and outbreak type (acute/slow). Variables were included in the model if their *p*-value as a single variable in the model was <0.10 and their inclusion improved the goodness of fit of the model. Models were assessed for the distribution of the residuals. The final model for duration included the number of interventions, the proportion of biters, the proportion of victims, and the interaction “last method × number of interventions”. The final model for intervention success included the first and the last method, age, the proportion of biters, and the proportion of victims.

The success of adding rope as an intervention was further analysed independently between trials using the same method as described above to assess if different enrichment backgrounds (i.e., the routine enrichment pigs were provided with on an ongoing basis during their lives) influenced the effectiveness of intervention by adding additional ropes.

Paired *t*-tests were used to compare the proportion of pigs with different lesion scores before and after the outbreak for normally distributed variables (proportion of pigs with moderate tail damage and blood score 2 & 3), and the Mann–Whitney test was used to compare non-normally distributed variables (proportion of pigs with tail damage or blood recorded at “score 0” and “score 3”).

Data are presented as least square means with standard errors unless otherwise indicated. 

## 3. Results

Over two trials, a total of 40 outbreaks were recorded in 34 pens. Although there were significantly more pens with tail biting outbreaks and longer outbreaks in trial 1 compared to trial 2 (*p* = 0.03, Table 2), the pattern of occurrence of tail biting outbreaks over time was similar between the trials (Figure 2). Figure 2 also shows that outbreaks started to occur about 11 days, and peaked around three weeks, after weaning. On average, there were 2.8 ± 0.3 biters and 7.7 ± 0.4 victims per pen of 12–14 pigs. The average proportion of biters (out of the total number of pigs in the pen) identified over the course of an outbreak per pen was 0.21 ± 0.02, and the proportion of victims was 0.58 ± 0.03. An average of 3.1 ± 1.7 biters was removed for “remove biters” interventions, and 4.9 ± 2.4 victims were removed for “remove victims” interventions. In total, five pigs were removed from the main study of trial 1, and four pigs (three in trial 1 and one in trial 2) were euthanised due to tail biting outbreak. Across the trials, six pens had a second recurring outbreak (Table 2) after the first had been resolved, and eight outbreaks were slow (Table 2), meaning that the outbreak lasted >72 h. 

### 3.1. Intervention Success

All three interventions were equally represented across the intervention steps (maximum three steps; Table 3).

Exactly half of the interventions (20/40) were successful after the first intervention was carried out. Ten outbreaks required a second intervention, and another ten outbreaks required a third intervention. Of these last ten, only two interventions were successful (i.e., eight out of 40 outbreaks were not successfully resolved after all three intervention strategies were applied). Survival analysis showed that if three interventions were used, there was an 80% predicted chance of the tail biting outbreak continuing (Figure 3a).

The success of the last method applied tended to be the lowest in the rope treatment (F_2,25_ = 2.55; *p* = 0.09), with the probability of success for removing the biter being 0.90 ± 0.10, for providing ropes 0.54 ± 0.14, and for removing the victim 0.91 ± 0.09. Success of the last used intervention was related to the proportion of biters per pen, with more biters increasing the chance of failure (b = 20.77 ± 8.35/0.1 increase in proportion of biters; F_1,25_ = 6.18; *p* = 0.02; Figure 4). Similarly, the failure was related to the proportion of victims in the pen (b = 10.10 ± 4.60/0.1 increase in proportion of victims; F_1,25_ = 4.82; *p* = 0.04; Figure 4). Specifically, Figure 4 shows that the likelihood to successfully overcome an outbreak is lower when more biters are in the pen as compared to having the same number of victims. The probability of success of the last method was greater when pigs were older (b = −0.626 ± 0.325/week of age; F_1,26_ = 3.71; *p* = 0.065). Trial, month of the year, and whether outbreaks were acute or slow were not significantly related to the success of the last method (*p* > 0.10). The success of the first intervention was not influenced by the three methods used either (*p* > 0.10).

### 3.2. Intervention Duration

The duration of an outbreak increased with the number of interventions needed to terminate it (b = 6.89 ± 2.786/intervention; F_1,22_ = 6.25; *p* < 0.001). When only one intervention was used, providing ropes resulted in a shorter outbreak duration than removing the victim (add ropes 9.17 ± 1.50 d v’s remove victims 15.78 ± 1.23 d, *p* = 0.01; Figure 3b) and tended to be shorter in duration compared to the strategy of removing the biter based on Kaplan–Meier survival analysis (remove biters 13.00 ± 1.65 d, X^2^ = 4.77, *p* = 0.05; Figure 3b). There was no difference in the duration of the outbreak between removing the biter versus removing the victim (*p* = 0.40). When there were two intervention strategies needed to overcome the outbreak, there was no difference in the duration of the outbreak based on the last strategies applied (*p* = 0.11). As single variables in the model, the proportion of biters and victims and the interaction between number of methods and the last method significantly influenced the duration, but these effects became non-significant when they were combined into the model with number of interventions (*p* > 0.10). Outbreak duration did not significantly differ between trials, months of the year, what the last method was, the age of the pigs, or whether the outbreak was acute or slow (*p* > 0.10).

### 3.3. Tail Lesion Scores

Detailed tail lesion scores were obtained as per the main studies’ protocols at a fortnightly basis. As a result, tail lesions were scored on average 6.9 ± 0.8 days before outbreaks began and 8.6 ± 1.0 days after outbreaks finished. The proportion of pigs with moderate or severe tail scores (score 2 and 3 combined) and severe tail scores (score 3) was lower when the outbreak was resolved compared to before the onset of the outbreak, both in terms of tail damage and blood presence (Table 4). Pigs with no tail damage (score 0) did not differ before or after the outbreak, but more pigs did not have blood on tails after the outbreak (Table 4). 

The proportion of pigs with a severe tail damage score (score 3) after outbreaks was affected by the proportion of victims in the pen only when included as a single variable in the model (b = 0.07 ± 0.03/0.1 increase in proportion of victims *p* = 0.04). There was no difference in the proportion of pigs with moderate damage or blood score (score 2 and 3 jointly), severe blood score (score 3), or no lesion (score 0) between numbers of intervention used, last methods, types of outbreak, proportion of biters present, and duration of outbreak.

## 4. Discussion

This study aimed to explore the outcome of three tail biting intervention methods in a multi-step protocol during tail biting outbreaks that were defined to reflect what happens on commercial farms, both in terms of the timing and the nature of the interventions. Half of the outbreaks were controlled with only one intervention, but 20% of the outbreaks could not be overcome, even when all three intervention methods were used. The proportion of biters and victims in the pen had a greater influence on the success rate of resolving a tail biting outbreak than the intervention methods used. The multi-step intervention used in the current study required all removed pigs to be reintroduced in the home pens before an outbreak was deemed resolved, and this could be more applicable to practice where there is limited amount of extra space.

### 4.1. Intervention Success and Duration

By using a multi-step intervention protocol, which employs different methods progressively to accommodate situations where one intervention method fails, it was possible to increase success from 50% after one intervention to 80% of the 40 outbreaks recorded. This was a positive outcome considering that our definition of a tail biting outbreak adopted a higher threshold of injury to the pigs than was the norm in previous research (three out of 12–14 pigs with severe tail damage) [14,15]. Under commercial settings, a multi-step protocol is realistic and necessary, since farmers need to resort to all possible measures to stop the outbreak from continuing.

There was no difference between intervention methods in terms of stopping the outbreak, which agrees with a previous study that compared removing biters and adding straw as intervention methods [14]. However, outbreaks were of shorter duration when ropes were added in comparison to removing animals. This was partly due to the time required for removed tail bitten victims to recover to a state that they could be reintroduced and the time required for reintroduction of the earlier removed pigs, which was not a component of the “additional ropes” treatment. Another disadvantage of removing biters and victims was the difficulty in identifying biters, given that biters could also be victims and hence have a double “role” [28,29]. It requires time and experience to identify the pig’s role through behavioural observations, which may not always be feasible on farms due to time restrictions. Moreover, a larger group size than what we had in this work (12–14 pigs) will make it more difficult to identify biters, especially if stocking density is high. This contributes to why some farmers are used to removing victims rather than biters [22,23]. Nevertheless, this study has shown that there was no difference in removing the victims or the biters, either in terms of intervention success rate or duration of the outbreak, and thus it may not have major practical implications whether the biter or the victim is removed. In this study with 48 groups of pigs, at maximum, ten hospital pens were needed to accommodate the removed pigs due to outbreaks without mixing different pens. In practice, pens could be divided into smaller pens to avoid the need for regrouping and to enable this intervention. Further work is still needed to investigate how group size may also influence the outcome of different intervention methods.

Adding in ropes reduced the outbreak duration, albeit there was no clear benefit with regard to the number of successful interventions. As previous studies suggested, ropes may not be the most effective enrichment material to stop a tail biting outbreak compared to chopped wheat straw [15]. On fully slatted floors, providing straw on the floor is not possible, and instead it could be more effective to supply loose substrates in suitable dispensers (e.g., hay racks) if placed in the correct location in the pen to avoid aggression [30].

The chance of resolving a tail biting outbreak was affected by the proportion of biters and victims identified in a pen. When more biters or victims were identified, the probability of having a successful intervention dropped. Tail biting outbreaks happen in a rapid and sudden fashion and can also spread from pig to pig [10]. The longer tail biting continues without intervention, the more pigs may become potential biters and victims. This also reiterates the importance of early detection for prompt intervention [15,17,18,31,32,33]. 

### 4.2. Intervention and Tail Lesion Scores

The tail lesion scores after outbreaks were resolved were lower than before, which confirmed the effectiveness of the interventions in overcoming the outbreaks. However, no difference of the tail scores was found between the three different intervention methods, the numbers of intervention used, or the duration of the outbreaks. Tail lesion scores were not scored on a daily basis or according to the onset of the outbreaks but every two weeks based on the schedule of the main studies. It is acknowledged that some tail lesions may have healed in the time between the outbreak and the recording day. The scores thus may not truly reflect the direct damage pattern within the outbreak. Previously, it was reported that the cutaneous healing of tail injuries appears to take place over 3–7 days [34], and therefore with more frequent tail lesion scoring (e.g., every 2–3 days), we may have detected differences between the intervention strategies. 

### 4.3. Reintroduction of Ex-Biters and Ex-Victims

By using a strict protocol in removing and reintroducing pigs, the majority of pigs removed due to intervening tail biting outbreaks were able to be reintroduced. To the best of our knowledge, this study is the first to report that routine reintroductions of previously removed pigs back to the home pens is feasible after tail biting outbreaks. Although no detailed recording of skin lesions was conducted, which is the common method to score aggression between pigs [35], no overt aggression or clearly visible skin lesions were observed during subsequent routine monitoring. Nevertheless, it cannot be ruled out that some level of aggression might have occurred without clear signs. In experimental settings as well as in practice, pigs that are taken out of the pen are hardly ever reintroduced or re-mixed due to the fear of aggression at regrouping [24]. Zonderland et al. (2008) removed biter pigs from their home pen and later regrouped them into new pens instead of reintroducing them into their original group [14]. Although this regrouping did not result in further tail biting, the consequences of mixing unfamiliar animals and precautionary measures taken to reduce subsequent aggression was not reported [14]. Moreover, newly formed groups needed to occupy additional space permanently. Caution is needed when remixing or re-introducing pigs, since aggression and the negative consequences of fighting due to remixing are well-documented [24], and pigs may even start fighting upon the introduction of a previous member [36]. Successful reintroduction (i.e., with limited aggression) of a previous group member may depend on previous hierarchy, with dominant animals successfully returning after 25 days of absence, whereas subordinate animals may be attacked even after being removed for three days [36]. However, there are techniques that can help mitigate aggression at mixing [37]. Our study therefore suggests that, with good management practices and accurate record keeping, it is possible to reintroduce pigs back into their home groups, which in turn can free up space in the hospital pens and reduce the requirement for extra space.

### 4.4. Different Types of Outbreaks

In the current study, both the success and the duration of the interventions were affected by the proportion of biters and victims present in the pen, which might have represented different “types” of outbreaks. Depending on the clinical signs and the development phase, tail biting has been categorised in a range of different types (e.g., two-stage, sudden forceful, obsessive, epidemic [8,29]). As the occurrence of tail biting may have different origins [5,6,8,29], intervention methods could be tailored to the type of outbreak in order to stop the outbreak efficiently. For example, in the case of an obsessive biter, it might be most effective to immediately remove the biter if it can be identified. To investigate this, more outbreak data are needed than we had available. It may be necessary to use meta-analysis, which would only be possible if classification of outbreak type were standardised across experiments. Thus, it would be meaningful to construct a classification tool to help both researchers and producers identify different types of outbreaks, with the aim of determining the most effective intervention.

### 4.5. Limitation in Data Collection

The outbreak data used in the current study were collected from two different trials that took place in the same research facility with the same breed of pigs and management practices. Although the two trials were conducted under very similar conditions, the difference in enrichment provision could make the pigs react differently to the ropes as additional enrichment. We recognise this may influence the outcome of the outbreak interventions; therefore, in order to ensure the differences between trials were accounted for, “trial” was always included in the statistical models. There was no difference in the success of intervening with ropes between the two trials. Indeed, it can be argued that using data from two different trials does increase the validity of the protocols and their applicability under a range condition. 

Furthermore, this study used a multi-step intervention protocol, which is beneficial for practical application, but this methodology also complicated the experimental design since it is impossible to predict in advance how many steps will be required before an outbreak is successfully controlled. The analysis of the intervention combinations would have benefitted from a larger number of outbreaks; however, over two trials, only 34 pens (out of 96, 35%) recorded a tail biting outbreak. Therefore, we were only able to best analyse the data by comparing the method that led to the success of the intervention (i.e., the last one). The limited number of outbreaks thus prevented a full interpretation of how different combinations and order of the methods used could have affected the success of the intervention when more than one intervention was used. Future studies with a larger scale commercial trial could explore this further. 

## 5. Conclusions

This study developed a multi-step intervention protocol to controlling tail biting outbreaks in pigs involving the removal of either (a) the biters or (b) the victims and (c) the provision of three ropes, which has practical application in pig production systems. Eighty percent of all tail biting outbreaks were controlled within three intervention steps outlined in the protocol. Compared within one-step interventions, adding ropes resulted in relatively shorter duration of outbreaks, but there was no difference in the success rate as a last treatment between the intervention methods used. The proportion of biters and victims in a pen had the greatest influence on the success of controlling an outbreak, rather than the intervention methods used. This emphasizes the need to act promptly at, or even before, the onset of an outbreak in order to increase the likelihood of successful control. It was possible to reintroduce all removed pigs back into their home groups, which is crucial to reduce extra space requirement. As removal of pigs without reintroduction implies that the underlying problem is still prevailing, reintroduction should be considered part of a successful intervention strategy.

## Figures and Tables

**Figure 1 animals-09-00582-f001:**
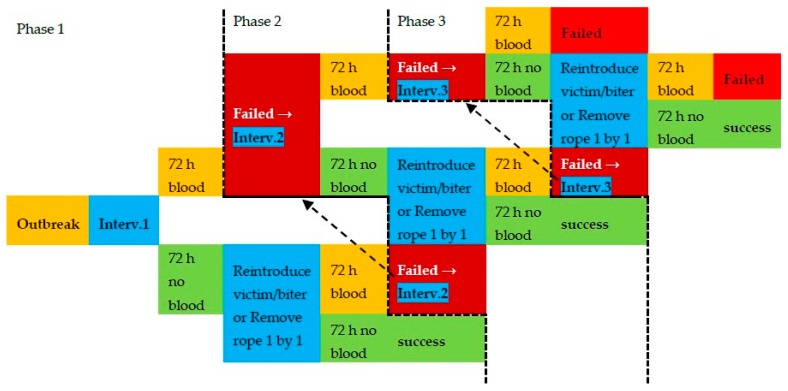
Protocol for the intervention (Interv.) steps after an outbreak and the classification of success and failure in Phase 1–3 (separated by dotted line). Colours indicate the action taken (blue) and the urgency of the outbreak (green = getting better/resolved and yellow/red = getting worse/could not be controlled).

**Figure 2 animals-09-00582-f002:**
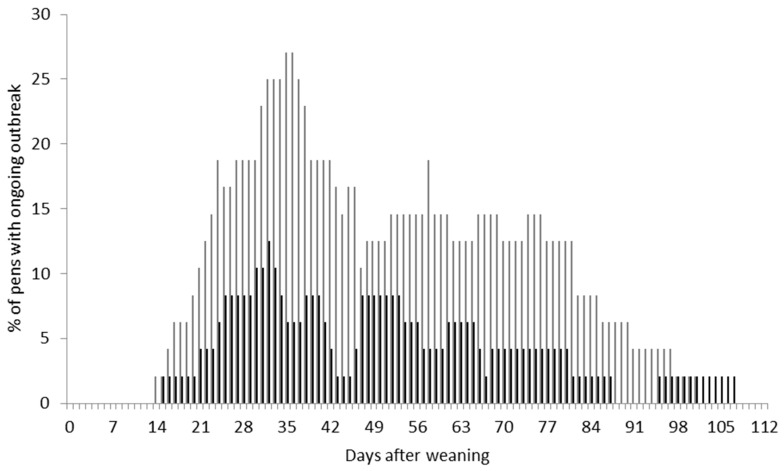
Percentage of pens with ongoing tail biting outbreaks (defined as being between the criteria for the start of an outbreak and successful resolution, criteria for each of these are explained in the text) plotted against days post-weaning within trial 1 (light grey) and trial 2 (black).

**Figure 3 animals-09-00582-f003:**
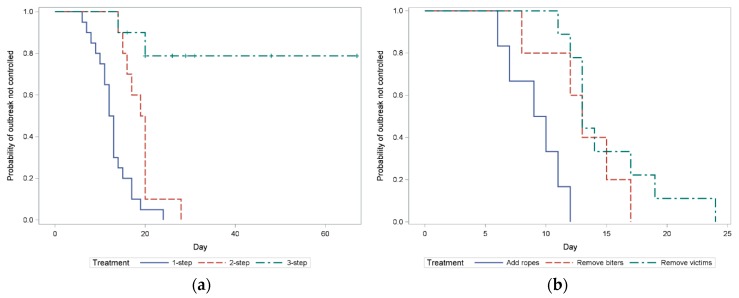
Kaplan–Meier survival plots for the probability (y-axis) of a tail biting outbreak continuing, where higher survival probability represented a higher chance that the tail biting outbreak was continuing and not controlled. (**a**) Number of interventions used, blue solid line: outbreaks with one intervention used, red dashed line: two interventions, and green dashed-dotted line: three interventions. (**b**) The outbreaks that were successfully controlled with one intervention: blue solid line: adding ropes, red dashed line: removing biters, green dashed-dotted line: removing victims.

**Figure 4 animals-09-00582-f004:**
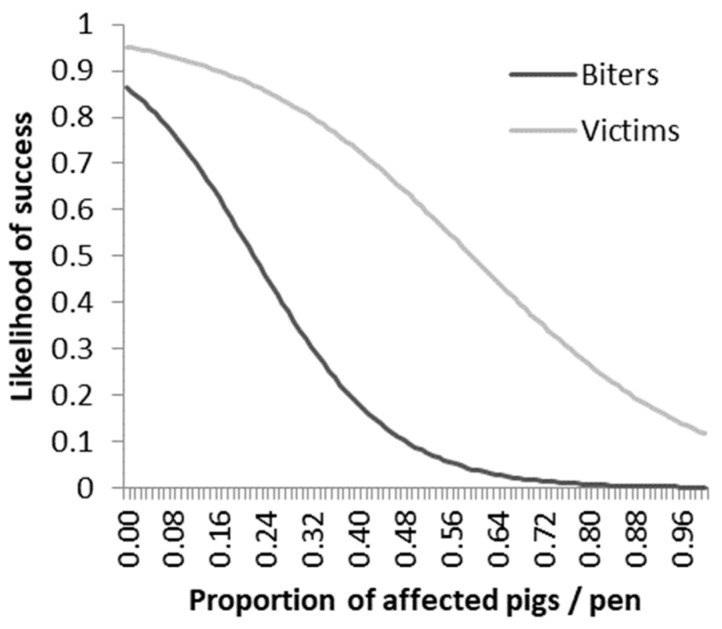
Probability of a successful intervention based on the proportion of biters and victims identified during the course of an outbreak in the pen. The curves are plotted using the intercepts and betas extracted from the logarithmic model of the data.

**Table 1 animals-09-00582-t001:** Schematic process of a successful one-step intervention.

Day	D0	D1	D2	D3	D4	D5	D6	D7	D8	D9	D10
Outbreak identified	Remove biters/+victims	Monitor	Monitor	Reintroduce + 3 ropes	Remove 1st rope	Remove 2nd rope	Remove 3rd rope	Monitor	Monitor	Monitor	Success
	Add ropes × 3			Remove 1st rope	Remove 2nd rope	Remove 3rd rope	Monitor	Monitor	Monitor	Success	

**Table 2 animals-09-00582-t002:** Differences in tail biting outbreaks between the two trials.

Comparisons	Trial 1	Trial 2	Test	*p*-Value
Pens with outbreaks	22	12	*X*^2^ (1, *N* = 96) = 4.55	0.03
Pens with recurring outbreaks	4 (18.2%)	2 (16.7%)	-	-
Pens with slow outbreaks (>72 h)	5 (19.2%)	3 (21.4%)	*X*^2^ (1, *N* = 40) = 0.03	0.87
Mean duration of outbreaks (d)	19.6	13.3	*t* (34) = 2.28	0.03
Successful interventions (%)	76.92	85.71	*X*^2^ (1, *N* = 40) = 0.44	0.51
Interventions used (median)	2	1	*U* (*N_1_* = 26, *N_2_* = 14) = 267	0.58

**Table 3 animals-09-00582-t003:** The number of successes and failures for each of three intervention methods (remove biter: B; remove victim: V; give ropes: R) over 3 steps (percentages shown in brackets).

Method	1st Step	Result	Count (Percentage)	2nd Step	Result	Count (Percentage)	3rd Step	Result	Count (Percentage)
B	14 (35.0%)	Fail	9 (22.5%)	7	Fail	3 (15.0%)	2	Fail	1 (10.0%)
		Success	5 (12.5%)		Success	4 (20.0%)		Success	1 (10.0%)
V	16 (40.0%)	Fail	7 (17.5%)	8	Fail	5 (25.0%)	2	Fail	1 (10.0%)
		Success	9 (22.5%)		Success	3 (15.0%)		Success	1 (10.0%)
R	10 (25.0%)	Fail	4 (10.0%)	5	Fail	2 (10.0%)	6	Fail	6 (60.0%)
		Success	6 (15.0%)		Success	3 (15.0%)		Success	0 (0.0%)
Total	40	Fail	20 (50.0%)	20	Fail	10 (50.0%)	10	Fail	8 (80.0%)
		Success	20 (50.0%)		Success	10 (50.0%)		Success	2 (20.0%)

**Table 4 animals-09-00582-t004:** Proportion of pigs (mean ± s.e.) with different levels of tail damage and blood presence scores recorded before the onset of outbreaks and after outbreaks were resolved.

**Tail Damage**	**Before ^1^**	**After ^2^**	**Test**	***p*-Value**
Score 0	0.11 ± 0.02	0.15 ± 0.03	*U* = 1115.0	0.481
Score 2 & 3	0.26 ± 0.03	0.10 ± 0.02	*t* = 4.00	< 0.001
Score 3	0.09 ± 0.02	0.02 ± 0.01	*U* = 1397.0	0.006
**Blood Presence**	**Before ^1^**	**After ^2^**	**Test**	***p*-Value**
Score 0	0.21 ± 0.03	0.35 ± 0.04	*t* = −2.77	0.007
Score 2 & 3	0.38 ± 0.04	0.16 ± 0.03	*t* = 4.67	< 0.001
Score 3	0.08 ± 0.02	0.03 ± 0.01	*U* = 1334.5	0.048

^1^ Score recorded on average 6.9 ± 0.8 days before outbreaks began. ^2^ Score recorded on average 8.6 ± 1.0 days after outbreaks finished.

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
