# Peer review of "Multi-Step Tail Biting Outbreak Intervention Protocols for Pigs Housed on Slatted Floors"

_animals, 2019, doi:10.3390/ani9080582_

Round 1

Reviewer 1 Report

The subject of prevention and control of tail biting outbreaks is of great topical interest at the current time and represents a significant gap in our scientific knowledge. This paper is therefore very important as one of the few large-scale controlled studies on the subject of intervention methods. This is a difficult subject to study, and the work here has been carefully planned and carried out with due ethical consideration. I would commend the authors on the introduction, which gives an excellent review of the current state of the art

Specific comments:

L174. Were these pigs allocated a hospital pen to themselves, or mixed with victims from other groups in a common hospital pen?

L180. Same question regarding a dedicated or common isolation pen for these animals.

[OK, just found this at line 228, but it would have been more helpful in the earlier location. Would biters and bitten pigs removed from the same group in 2-phase interventions be housed together in isolation pens?]

L186. What was the function of this ointment. Did it contain antibiotic to prevent infection or an analgesic for damaged tails of pigs remaining in the pen?

L196. Was the second intervention applied immediately fresh blood was again seen, i.e. as quickly as 3h later if seen at the late afternoon inspection.

L246. If this meant removal of a single bitten pig, presumably according to the criteria used in the main study, how was this pig managed subsequently? Was it later reintroduced? It would be useful to know how frequently pig removal without a full outbreak occurred.

L294. Just to be clear, this is the proportion of pigs in the pen in each of these categories, not an implication that others in the pen existed but were ‘unidentified’. It might be better to say something like ‘proportion of pigs in the pen classified as…’. However, were biters identified and recorded in cases where the intervention designated was V or R?

L297. What is meant by ‘dropped’ were these pigs removed in the absence of an outbreak?

Fig 3. Since up to 25% of pens has simultaneous outbreaks this must have meant a lot of hospital/isolation pens were needed – something for the discussion.

Fig 4. I am unclear how to interpret this. By definition, should all ‘successful’ one phase outbreaks not be completed in 9 (rope) or 10 (removal/reintroduction) days as shown in fig 2.

L328. Does this take account of the fact that the rope was used more often than other interventions as the third treatment?

Fig 5. Should both lines not have an intercept of 1.0 for 0 pigs?

L349. Was this just because the defined ‘time to success’ (Fig 2) was one day less?

L361. Do these timings refer to the first and last day of any pig observed with a non-0 score?

[reading later in the discussion, I see that these data come from the fortnightly inspections so were these the average times of the last inspection date before the outbreak start and the first after a defined success? It would be useful to make this clearer here]

L388 or 459. But is the number of separate pens required in this study really commercially feasible?

L401. This implies that removed pigs were not always reintroduced after 72h in a successful intervention. I don’t think this was previously stated in methods (unless I missed it) – what were the criteria for delaying reintroduction?

L408. ..rather than biters?

L442. What is defined as ‘excessive’?

L488. But is not comparison of the success of the first intervention method a less confounded result?

L498. Make it clear if this is success rate at first intervention or across all interventions.

Author Response

Reviewer 1

The subject of prevention and control of tail biting outbreaks is of great topical interest at the current time and represents a significant gap in our scientific knowledge. This paper is therefore very important as one of the few large-scale controlled studies on the subject of intervention methods. This is a difficult subject to study, and the work here has been carefully planned and carried out with due ethical consideration. I would commend the authors on the introduction, which gives an excellent review of the current state of the art

Response: Dear reviewer, thank you for your appreciation of this study. We have incorporated each of your suggestions, detailed below, in the revised version.

Specific comments:

L174. Were these pigs allocated a hospital pen to themselves, or mixed with victims from other groups in a common hospital pen?

Response: They had a hospital pen to themselves (added in L180-181).

L180. Same question regarding a dedicated or common isolation pen for these animals.

[OK, just found this at line 228, but it would have been more helpful in the earlier location. Would biters and bitten pigs removed from the same group in 2-phase interventions be housed together in isolation pens?]

Response: there was only one intervention method in use at a given time, and therefore the biters and the victims were not removed at the same time point (added in L187-188).

L186. What was the function of this ointment. Did it contain antibiotic to prevent infection or an analgesic for damaged tails of pigs remaining in the pen?

Response: It is a soothing ointment containing mainly plant extracts for dermatological wounds without antibiotic. It also contained Neem oil with a special odour and based on our previous experience we found that pigs are not attracted to this odour and therefore it can cover the scent of the blood.

L196. Was the second intervention applied immediately fresh blood was again seen, i.e. as quickly as 3h later if seen at the late afternoon inspection.

Response: Yes, the second intervention was applied if fresh blood was again seen during the next inspection by the experimenter.

L246. If this meant removal of a single bitten pig, presumably according to the criteria used in the main study, how was this pig managed subsequently? Was it later reintroduced? It would be useful to know how frequently pig removal without a full outbreak occurred.

Response: Pig removal unrelated to a tail biting outbreak was handled in the same way as the reintroduction protocol (always with companion, reintroduced within 14 d, etc.). Due to the focus of this study on tail biting outbreaks, the removal of other pigs has not been mentioned here but are reported in the publication of the main studies (added in L251-252).

L294. Just to be clear, this is the proportion of pigs in the pen in each of these categories, not an implication that others in the pen existed but were ‘unidentified’. It might be better to say something like ‘proportion of pigs in the pen classified as…’. However, were biters identified and recorded in cases where the intervention designated was V or R?

Response: This is the proportion of biters/victims out of the number of pigs per pen. Biters/victims were always recorded in detail during outbreaks/routine health checks (added in L300).

L297. What is meant by ‘dropped’ were these pigs removed in the absence of an outbreak?

Response: L303-304 amended

Fig 3. Since up to 25% of pens has simultaneous outbreaks this must have meant a lot of hospital/isolation pens were needed – something for the discussion.

Response: Although up to 25% of pens have simultaneous outbreaks (only in trial 1), not all pens were removing animals at the same time, and an estimated 10 hospital pens were at maximum used at a single time point for 48 pens. In practice, a hospital pen could be split up with barriers to create several smaller pens since only a small number of pigs have to be removed together per pen. Investing in several smaller pens to enable this protocol would possible be cost beneficial in comparison to the costs that can rapidly accumulate from tail bitten pigs. This has been added to the discussion (L421-424).

Fig 4. I am unclear how to interpret this. By definition, should all ‘successful’ one phase outbreaks not be completed in 9 (rope) or 10 (removal/reintroduction) days as shown in fig 2.

Response: Fig 2 is based on theory, whereby victims recover within 3 days. Fig 4 is based on practice, whereby there is a delay due to victims first needing to have their tails healed before being reintroduced (added in L249-250).  

L328. Does this take account of the fact that the rope was used more often than other interventions as the third treatment?

Response: Luckily, 50% of the outbreaks could be resolved with one treatment only. As a consequence there were only very few cases (n=10) that needed 3 different interventions and as we only could randomise the first intervention, and it was impossible to know in advance how many outbreaks would progress into the third stage, which resulted in more ‘rope interventions’ being by chance as the third intervention. The number in this category is too low to assess whether this is by chance or a pattern. However, the results of the immediate successful intervention indicate that there was no difference in the success of the three interventions.

Fig 5. Should both lines not have an intercept of 1.0 for 0 pigs?

Response: This logarithmic figure is based on the intercept and beta from the statistical model, modelling y = A (interc.) + B (slope) * X (proportion starting at 0). As the model takes into account other factors as well, the intercept is not zero. Setting the intercept manually to start at a likelihood of 1 would be falsely adjusting the subsequent values.

L349. Was this just because the defined ‘time to success’ (Fig 2) was one day less?

Response: As shown in Fig 4b, the difference is more than 1 day (as would be the a priori difference). The results from the mixed model showed that the difference was greater than 1 day (add ropes 9.17 ± 1.50, remove biters 13.00 ± 1.65, remove victims 15.78 ± 1.23 d, P = 0.02) (L357-360 added)

L361. Do these timings refer to the first and last day of any pig observed with a non-0 score?

 [reading later in the discussion, I see that these data come from the fortnightly inspections so were these the average times of the last inspection date before the outbreak start and the first after a defined success? It would be useful to make this clearer here]

Response: We have mentioned the timing of lesion scoring in section 2.2 L164-165, and added it in addition in L370-371.

L388 or 459. But is the number of separate pens required in this study really commercially feasible?

Response: As previously responded and also discussed in L421-424.

L401. This implies that removed pigs were not always reintroduced after 72h in a successful intervention. I don’t think this was previously stated in methods (unless I missed it) – what were the criteria for delaying reintroduction?

Response: Added in L249-250.

L408. ..rather than biters?

Response: L418 amended.

L442. What is defined as ‘excessive’?

Response: Rephrased to ‘clearly visible’ (L455)

L488. But is not comparison of the success of the first intervention method a less confounded result?

Response: As a whole intervention protocol, we think the first method would not reflect the success of the whole intervention protocol compared to the last/immediate intervention used within the protocol, considering that not all interventions were overcome by the first method. However, we do agree with the reviewer that the success of the first intervention would be also interesting in telling us more about the different intervention methods used in the study, and therefore we ran the analyses and included relevant methods (L275 & L283) and results (L346-347).

L498. Make it clear if this is success rate at first intervention or across all interventions.

Response: Amended accordingly (L510-511).

Reviewer 2 Report

General comment: The manuscript is well structured and written, it is clear and very interesting. Furthermore, the experiment provides important practical suggestions regarding the reintroduction of victims/biters which are of great interest from a practical standpoint.

Here some minor comments:

Lines 23-24: At first glance it may seem that the three interventions are always in sequence (removing of the biter first, removing of the bitten per second and use of ropes per third). I would suggest specifying, even in the simple summary, that the different interventions were randomly assigned.

Line 47: I personally have some doubts about the fact that the term "intact tail" is a useful key word. Could it perhaps be better to use the word "undocked"?

Line 135: in the Animals and Housing section, please specify the duration of the illumination period due to its impact on animals’  behavior. For the same reason,  information should be provided concerning diet formulations (energy, protein and fibre supplies).

Line 155: the FareWellDock scoring system should be, in my view, quoted as such, i.e. from the site of the FareWellDock project itself, and not as Chou et al., 2019 (corresponding to reference n.27). This same quotation should be repeated throughout the text whenever reference is made to FareWellDock (e.g. lines 162, 164).

Lines 173: the fact that victims were removed in at least threes, might create some confusion with what stated in line 230 (at least two pigs were removed together). I wonder if it is so necessary to specify that the victims have been removed in groups of three…

Line 232: The calculation through which the value of 14 d is obtained should be better explained (neither figure 1 or figure 2 can help me in this sense).

Figure 2: Please check the format. Have the pigs that received the ropes been monitored on D1 and D2 (the corresponding boxes appear empty)? When ropes are used as the first intervention, is the success achieved a day early? Is something missing?

Figure 4: the different colors corresponding to the different lines are not well distinguishable.

Author Response

Reviewer 2

General comment: The manuscript is well structured and written, it is clear and very interesting. Furthermore, the experiment provides important practical suggestions regarding the reintroduction of victims/biters which are of great interest from a practical standpoint.

Response: Dear reviewer, thank you for your recognition. We have amended the manuscript based on your comments, detailed below, in the revised version.

Here some minor comments:

Lines 23-24: At first glance it may seem that the three interventions are always in sequence (removing of the biter first, removing of the bitten per second and use of ropes per third). I would suggest specifying, even in the simple summary, that the different interventions were randomly assigned.

Response: L23-24 amended accordingly.

Line 47: I personally have some doubts about the fact that the term "intact tail" is a useful key word. Could it perhaps be better to use the word "undocked"?

Response: L48 keywords amended accordingly

Line 135: in the Animals and Housing section, please specify the duration of the illumination period due to its impact on animals’  behavior. For the same reason,  information should be provided concerning diet formulations (energy, protein and fibre supplies).

Response: L147-148 and L150-153 information added accordingly

Line 155: the FareWellDock scoring system should be, in my view, quoted as such, i.e. from the site of the FareWellDock project itself, and not as Chou et al., 2019 (corresponding to reference n.27). This same quotation should be repeated throughout the text whenever reference is made to FareWellDock (e.g. lines 162, 164).

Response: We agree with the reviewer’s suggestion, but when the study took place, the system at that time was a previous version. Therefore, if the current FareWellDock website/system is referenced, it will lead to the new system which was not present at the time of the study. Therefore, we cited our earlier study which had a more detailed description of the actual system used as an alternative.

Lines 173: the fact that victims were removed in at least threes, might create some confusion with what stated in line 230 (at least two pigs were removed together). I wonder if it is so necessary to specify that the victims have been removed in groups of three…

Response: The sentence deleted accordingly (before L180).

Line 232: The calculation through which the value of 14 d is obtained should be better explained (neither figure 1 or figure 2 can help me in this sense).

Response: The 14d of removal is a cut-off point we decided in advance to ensure minimum issue during reintroduction. It is based on past literature and experience so it wasn’t calculated. (L235-236 added accordingly)

Figure 2: Please check the format. Have the pigs that received the ropes been monitored on D1 and D2 (the corresponding boxes appear empty)? When ropes are used as the first intervention, is the success achieved a day early? Is something missing?

Response: We apologise that the format of Figure 2 was altered in the submitted manuscript, and now it was corrected and revised to make it more readable. The pigs were monitored on D0, D1 and D2 as in the remove animal treatment (the same in the rope treatment). There was a one-day difference between remove animal and add rope but it is an unavoidable difference due to the reintroduction of the animals.

Figure 4: the different colors corresponding to the different lines are not well distinguishable.

Response: We will upload a full resolution file separately as it will be clearer than the one attached to the main text, and also adjust the lines thicker now.

Reviewer 3 Report

The reviewed manuscript deals with a major concern in pig industry which is the unpredictable and costly occurence of tail biting outbreaks. The subject area is interesting, especially with regard to the necessity to find cost-effective intervention strategies by employing a standardized protocol. The manuscript is very well-written and of high quality, in both content and elaboration while the results are clear and of practical interest. Therefore, I can only suggest some minor modifications before publication:

P3, L 140-146: Were groups mixed/re-graded between production stages? I would suggest to include 1 or 2 lines to provide some details on how pigs were managed on farm. The authors only say that ‘at 11 weeks of age, the pigs were transferred to the finisher housing.’ P4, L 146: Please, remove the extra space between ‘building’ and ‘and’. P4, L 150-152: The authors say that the health checks was made every day by the experimenters at 3 specific time points and during other time points (please specify when, if possible) by the farm staff. Is there a motivation behind these two types of inspection? P6, L 232: The authors state that one of the rules applied to facilitate animals’ reintroduction was that ‘pigs were returned within 14 days of removal’. I suppose that the case in which all of 3 interventions were applied by adding 3 x 72h monitoring (i.e. 9 days) to the day of the measures taken (i.e. 3 days) which leads to a total of 12 days. Can you please clarify a situation of when pigs were returned at day 14 of removal? I am not sure I understood correctly. P7, L 290-298: Please be consistent with the terminology as you first refer to ‘trial’ (See L 290) and later in the manuscript at L 297 you use the word ‘exp.’ The same occurs in Table 1. I would suggest to check the text and amend it when needed. P9, Figure 4: I recommend the authors to upload a new version of Figure 4 with a better resolution. The figure, at the moment, has a poor quality and both numbers and text are not fully visible. I also found difficult to see the colours of the lines. P9, L 336-337: I would suggest to be consistent in presenting the values of means ± SE/P-value in L 336-337 as, for instance, those presented in L 331-332.

Author Response

Reviewer 3

The reviewed manuscript deals with a major concern in pig industry which is the unpredictable and costly occurence of tail biting outbreaks. The subject area is interesting, especially with regard to the necessity to find cost-effective intervention strategies by employing a standardized protocol. The manuscript is very well-written and of high quality, in both content and elaboration while the results are clear and of practical interest. Therefore, I can only suggest some minor modifications before publication:

Response: Dear reviewer, thank you for your kind words. We have incorporated your suggestions, detailed below, in the revised version.

P3, L 140-146: Were groups mixed/re-graded between production stages? I would suggest to include 1 or 2 lines to provide some details on how pigs were managed on farm. The authors only say that ‘at 11 weeks of age, the pigs were transferred to the finisher housing.’

Response: L153 amended accordingly

P4, L 146: Please, remove the extra space between ‘building’ and ‘and’.

Response: Done.

P4, L 150-152: The authors say that the health checks was made every day by the experimenters at 3 specific time points and during other time points (please specify when, if possible) by the farm staff. Is there a motivation behind these two types of inspection?

Response: L159 more information added. The routine checks were done by the farm staff as a daily practice on farm while the experimenter was checking the pigs and also taking experimental measures according to the experimental protocol of the main studies.

P6, L 232: The authors state that one of the rules applied to facilitate animals’ reintroduction was that ‘pigs were returned within 14 days of removal’. I suppose that the case in which all of 3 interventions were applied by adding 3 x 72h monitoring (i.e. 9 days) to the day of the measures taken (i.e. 3 days) which leads to a total of 12 days. Can you please clarify a situation of when pigs were returned at day 14 of removal? I am not sure I understood correctly.

Response: More information added in L249-250. As discussed in L409-412, there were complications that when the animals were not recovering to a state that allowed the reintroduction, due to ethical concerns, the reintroduction could be delayed. We think this is an important part to discuss since this is one risk of removing animals from pens and the possible prolonged time of reintroduction should be considered. Moreover, when an outbreak was not able to be controlled within the multi-step protocol and deemed failed, all measures would be taken to stop the continuance of tail biting, and during this time, the removal time of the animals could also be prolonged.

P7, L 290-298: Please be consistent with the terminology as you first refer to ‘trial’ (See L 290) and later in the manuscript at L 297 you use the word ‘exp.’ The same occurs in Table 1. I would suggest to check the text and amend it when needed.

Response: L303-304 and Table 1 amended accordingly

P9, Figure 4: I recommend the authors to upload a new version of Figure 4 with a better resolution. The figure, at the moment, has a poor quality and both numbers and text are not fully visible. I also found difficult to see the colours of the lines.

Response: We thank the reviewer of this recommendation and will upload a full resolution file separately as it will be clearer than the one attached to the main text, and also the lines are made thicker now.

P9, L 336-337: I would suggest to be consistent in presenting the values of means ± SE/P-value in L 336-337 as, for instance, those presented in L 331-332. 

Response: The difference in this data presentation is because the data in L339-341 are lsmeans with SEM for each category, whereas those in L344-345 are beta estimates for continuous (non-categorical) values.
